# Combinatorial Cu-Ni Alloy Thin-Film Catalysts for Layer Number Control in Chemical Vapor-Deposited Graphene

**DOI:** 10.3390/nano12091553

**Published:** 2022-05-04

**Authors:** Sumeer R. Khanna, Michael G. Stanford, Ivan V. Vlassiouk, Philip D. Rack

**Affiliations:** 1Department of Materials Science and Engineering, University of Tennessee, Knoxville, TN 37996, USA; skhanna@utk.edu; 2General Graphene Corporation, Knoxville, TN 37932, USA; mstanford@generalgraphenecorp.com; 3Oak Ridge National Laboratory, Oak Ridge, TN 37831, USA; vlassioukiv@ornl.gov

**Keywords:** thin film, combinatorial sputtering, alloys, catalyst, graphene, 2D materials, chemical vapor deposition (CVD), Raman spectroscopy

## Abstract

We synthesized a combinatorial library of Cu*_x_*Ni_1−*x*_ alloy thin films via co-sputtering from Cu and Ni targets to catalyze graphene chemical vapor deposition. The alloy morphology, composition, and microstructure were characterized via scanning electron microscopy (SEM), energy dispersive x-ray spectroscopy (EDS), and X-ray diffraction (XRD), respectively. Subsequently, the Cu*_x_*Ni_1−*x*_ alloy thin films were used to grow graphene in a CH_4_-Ar-H_2_ ambient at atmospheric pressure. The underlying rationale is to adjust the Cu*_x_*Ni_1−*x*_ composition to control the graphene. Energy dispersive x-ray spectroscopy (EDS) analysis revealed that a continuous gradient of Cu*_x_*Ni_1−*x*_ (25 at. % < *x* < 83 at.%) was initially achieved across the 100 mm diameter substrate (~0.9%/mm composition gradient). The XRD spectra confirmed a solid solution was realized and the face-centered cubic lattice parameter varied from ~3.52 to 3.58 $\dot{A}$, consistent with the measured composition gradient, assuming Vegard’s law. Optical microscopy and Raman analysis of the graphene layers suggest single layer growth occurs with *x* > 69 at.%, bilayer growth dominates from 48 at.% < *x* < 69 at.%, and multilayer (≥3) growth occurs for *x* < 48 at.%, where *x* is the Cu concentration. Finally, a large area of bi-layer graphene was grown via a Cu*_x_*Ni_1−*x*_ catalyst with optimized catalyst composition and growth temperature.

## 1. Introduction

The exceptional properties of graphene such as its good carrier mobility, optical properties (transparent), thermal conductivity, and carrier-density make it an ideal two-dimensional (2-D) material for various electronic, optical, and sensor applications [1,2,3]. The zero bandgap and semimetal nature of monolayer graphene, however, restricts its usage in electronic and optical applications. However, in AB-stacked bilayer graphene, the bandgap can be modulated, thus providing added potential for use in photonic and electronic devices [4,5,6]. Therefore, to make bi-layer graphene available for aforesaid applications, its production on large scale basis is necessary. 

Metal catalysts perform an essential role in governing the graphene growth quality, domain size, rate of production, and number of graphene layers [7,8,9,10]. Commonly, copper (Cu) and nickel (Ni) foils [11,12,13] are employed for graphene growth by chemical vapor deposition (CVD). Due to the self-limiting nature of CVD growth on copper (Cu) foils arising from low C solubility in Cu, monolayer graphene is predominantly formed [7,8,9,10,11,12,13]. However, there are some reports [14,15] to overcome this by adjusting the H_2_/CH_4_ ratio in low-pressure CVD processes. Bilayer graphene flakes produced by mechanical exfoliation and transfer process from bulk crystals yield domain sizes which are generally tens of micrometers [16]; however, they are accompanied by additional other variable thickness flakes. The methods tried in the past for bilayer graphene growth show good initial realization; however, 100% AB-stacked bilayer graphene over centimeter or greater dimensions is lately achieved [17]. 

Recently, bi-layer and multi-layer CVD graphene growth on Cu*_x_*Ni_1−*x*_ alloy has been reported [17,18,19,20,21]. The similar atomic size and crystal structure (face-centered cubic) of Cu and Ni metals (neighboring elements in the periodic table) favor the complete solid solubility in the alloy and thus a means of controlling the solubility of carbon in the solid solution at higher temperatures (~1000 °C). The solubility of C in Cu is very low (~75 ± 0.5 ppm) at 1000 °C and that of C in Ni is comparatively higher at 1000 °C (~1.3 at.%) [17]. Hence, by using a Cu*_x_*Ni_1−*x*_ alloy of appropriate composition one can regulate the graphene layer numbers during CVD graphene growth. Mechanistically it has been shown that the underlying growth mechanism for CVD graphene includes dehydrogenation reaction of the gaseous source of carbon precursors (CH_4_), diffusion of molecules (substrate), dissolution in Cu*_x_*Ni_1−*x*_ alloy solid solution, and precipitation of carbon atoms (supersaturation) at defect locations such as grain boundaries on the surface (equilibrium graphene growth) upon cooling [8,9,10,11,12,13,14,15,16,17,18,19,20,21,22,23] due to reduced solubility at low temperatures.

While there have been numerous studies using CuNi alloys to grow multilayer graphene, large area coverage of high-quality bi-layer graphene remains a challenge [7,8,9,10,11,12,13,14,15,16,17,18,19,20,21], due to the fine control of carbon solubility needed. The effect of Cu*_x_*Ni_1−*x*_ alloy composition on the number of layers of graphene by thermal CVD process has been studied including bi-layer Cu/Ni and Cu-Ni solid solutions [18,19,20,21]. Recently, large-area bilayer graphene has been synthesized by generating a series of custom CuNi catalyst foils with discrete compositions which enabled the precise carbon solubility to achieve bilayer graphene [17]. Although this approach yielded impressive bilayer graphene deposition, it is an arduous process to make custom CuNi foils in order to prototype CVD conditions which yield graphene of desired thickness. Additionally, the inter-diffusion of Cu/Ni layers and the loss of Cu at elevated temperatures can affect the catalyst composition and morphology during the CVD synthesis. Hence, it is desirable to have a catalyst alloy with a continuous gradient composition (rather than discrete composition foils) in order to rapidly prototype conditions for graphene grown of desired layer thickness.

In this study, we experimented with regulating CVD graphene layers and investigating the appropriate at.% of Cu*_x_*Ni_1−*x*_ alloy range for achieving predominantly monolayer, bilayer, and multilayer (≥3 L) CVD graphene. RF magnetron sputtering of a thin-film Cu*_x_*Ni_1−*x*_ gradient alloy catalyst (~2 μm thick) on Si/SiO_2_ substrate was performed. Graphene was subsequently grown on the gradient catalyst by an atmospheric pressure CVD process. By employing Raman spectroscopy, we showed that the gradient catalyst is indeed effective at growing a gradient in graphene thickness during a single CVD growth and therefore can serve as a method to rapid prototype recipe/catalyst combinations to grow graphene of controllable thickness. The resultant graphene grown on the gradient catalyst enabled compositions to be identified which yield predominantly monolayer (>85% Cu), bilayer (~61% Cu), and multilayer coverage (<49% Cu). Finally, we demonstrated a final proof-of-concept that combinatorial sputtering can be used to sputter a catalyst alloy of uniform composition which is optimized to produce large-area bilayer graphene. 

## 2. Materials and Methods

### 2.1. The Materials Thin-Film Sputtering of Cu_x_Ni_1−x_ Alloy Catalyst 

Thin-film (~1–2 µm thick) Cu*_x_*Ni_1−*x*_ alloy gradients were grown on silicon dioxide (SiO_2_)/silicon (Si) substrates via a combinatorial co-sputtering process [20,21,24]. The system was pumped to a base pressure of 3 × 10^−7^ Torr and backfilled with Ar to 5 mTorr. The 2-inch diameter targets were mounted and tilted such that when the substrate was not rotated, a composition gradient was achieved across the 100 mm diameter substrate. To achieve an initial composition Cu*_x_*Ni_1−*x*_ gradient of 25 at.% < *x* < 83 at.%, elemental Cu and Ni targets were first powered to 115 and 200 W, respectively, and sputtered for 66 min to achieve a ~1 μm thick film. A second, Cu-rich sample where 42 at.% < *x* < 94 at.% was co-sputtered with Cu and Ni sputtering powers 230 and 200 W, respectively, for 89 min to achieve a ~2 μm thick film. Finally, to investigate large-area bilayer graphene growth, a uniform ~2 μm (rotated substrate to eliminate composition gradient) Cu_63_Ni_37_ as catalyst film was co-sputtered for 180 min using Cu and Ni powers of 115 and 120 W, respectively. 

### 2.2. Energy Dispersive X-ray Spectroscopy and Scanning Electron Microscopy

The Cu*_x_*Ni_1−*x*_ composition as a function of position on the substrate was determined via energy-dispersive X-ray spectroscopy (EDS). EDS was measured using an X-ray detector mounted on a Carl Zeiss EVO scanning electron microscope (SEM) (Oberkochen, Germany) and spectra were obtained at 10 keV and a magnification of 10 kx. A Zeiss Auriga SEM was used to capture high magnification images of the Cu*_x_*Ni_1−*x*_ alloy catalyst before and after graphene growth. 

### 2.3. X-ray Diffraction

X-ray diffraction (XRD) (X’Pert Empyrean3 diffractometer from Malvern Panalytical, UK) of the as-deposited catalyst was measured at various positions on the substrate to correlate with the crystal structure of the catalyst and confirm the solid solution of the Cu*_x_*Ni_1−*x*_ alloy catalyst. An Empyrean X’Pert^3^ materials research diffractometer (MRD) was used with the following instrument parameters: wavelength Cu K_α1_ ($\dot{A}$) = 1.540598, K_α2_ ($\dot{A}$) = 1.544426, scan range (°): 38–99.98, start position = 38.03°, end position (°) = 99.95°, step size: 0.06°, number of points: 1033, omega = 12°, phi = 0° and chi = 0°.

### 2.4. Graphene CVD Growth and Transfer Process

All CVD experiments were conducted in a 3-zone furnace equipped with a 6” diameter quartz tube. The combinatorial CuNi catalysts sputtered on SiO_2_/Si wafers were placed into the furnace on a quartz pallet. The ends of the CVD tube were sealed by flanges and cooled by convection of ambient air. Graphene growth was achieved by flowing non-flammable stock gas mixtures including 5% CH_4_ in Ar and 2.5% H_2_ in Ar. The growth was conducted with the following steps: the first was a flow of 5 L/min of 2.5% H_2_ in Ar to purge the quartz tube. While flowing 5 L/min of 2.5% H_2_ in Ar, all three zones of the furnace were ramped to 1000 °C to anneal the catalyst for 1 h. Next, the carbon precursor was introduced by flowing 0.07 L/min and 0.25 L/min CH_4_ in Ar for all subsequent growths with the co-flow of 5 L/min of 2.5% H_2_ in Ar. After 30 min growth in the co-flow of 0.25 L/min CH_4_ in Ar with 5 L/min of 2.5% H_2_ in Ar, the heating elements were powered down and the furnace was opened to rapidly cool the samples. Co-flow of H_2_ and CH_4_ in Ar was maintained during cooling. 

Graphene was grown on the CuNi catalyst which was on a SiO_2_/Si wafer. For transfer, the graphene was spin-coated with a mixture of PMMA/anisole at 2000 rpm. The PMMA was cured at 150 °C on a hot plate. The CuNi catalyst was etched away by suspending the PMMA/Graphene/CuNi/Si samples on the surface of an aqueous (NH_4_)_2_S_2_O_8_ etchant. Surface tension caused the samples to float on the surface of the etchant and the CuNi etch slowly progressed from the edge of the samples. After the CuNi layer was etched away, the Si wafer fell to the bottom of the etching solution and left the PMMA/graphene floating on the surface. The graphene/PMMA was then rinsed in three water baths to remove residual etchant. The graphene/PMMA was then scooped onto a 300 nm SiO_2_/Si wafer. After water dried from the graphene/Si wafer interface, the sample was submerged in acetone to dissolve the PMMA film and rinse in IPA.

### 2.5. Raman Spectroscopy and Optical Microscopy

The growth quality of graphene layers is determined by performing Raman spectroscopy measurements. Raman spectra were obtained on a Renishaw inVia Qontor Raman microscope (Gloucestershire, UK) using a 532 nm laser for excitation. Raman mapping was constructed with Wire V5.0 software. Optical images were collected in a Keyence VHX 7000 series microscope. 

## 3. Results

### 3.1. Preliminary Evaluation of CVD Graphene Growth 

For a fixed gas composition, pressure, and temperature, the variance in the number of graphene layers grown on the Cu*_x_*Ni_1−*x*_ alloy is a function of the composition of Cu*_x_*Ni_1−*x*_ alloy. The focus here was to determine compositions of this alloy that provides the most uniform and coherent single, bilayer, and multilayer graphene growth. The rationale was to adjust the catalyst alloy’s carbon solubility such that the graphene layer number varies at ~1000 °C growth. The combinatorial co-sputtering process in which a wide composition space of the Cu*_x_*Ni_1−*x*_ alloy is obtained is schematically illustrated (Figure 1a). While a lateral gradient is imposed based on the geometry of the sputtering system, the Cu and Ni are uniformly distributed in the thickness as the sputtering conditions are not varied with time and thus the fluxes from the two targets are expected to be constant. An image of sputtered CuNi alloy sample showing a gradient in color from silvery-white Ni (left) to reddish-brown Cu (right) is illustrated (inset of Figure 1a). As shown, 5 positions were characterized via energy-dispersive X-ray spectroscopy, X-ray diffraction, and scanning electron microscopy. The resultant EDS spectra of the 5 positions measured every 16.6 mm on the sample where the Ni peak is observed at ~0.858 eV and Cu peak at ~0.937 eV, which correspond to the L_α_ values for both metals (Figure 1b). A plot of resultant atomic concentrations of Cu and Ni as a function of position across substrate is shown (Figure 1c). Glancing angle x-ray diffraction (GI-XRD) scans (Figure 1d) for the same five substrate positions showed a clear FCC structure with peaks shifting towards larger 2θ values with increasing nickel concentration indicative of the solid solution (inset of Figure 1d show (111) as densely packed plane of atoms, thus confirming FCC-type lattice structure of CuNi solid solution). Based on the peak positions, the calculated lattice parameters were calculated to be ~3.53, 3.54, 3.55, 3.57, to 3.58 $\dot{A}$. Consistent with the Vegard’s law, the composition of the solid solution can be calculated by a simple atomic fraction, which has been calculated and is in good agreement with the experimental EDS results. Finally, HR-SEM images of the as-deposited catalyst at the 5 substrate positions (Figure 1e) interestingly shows that the Ni-rich film is denser and has larger grains (disordered nodules) than the Cu-rich films which has the expected zone 1 voided columnar structure (uniform nodules/voided spheres) anticipated from a room temperature and low pressure sputtered film [21]. 

To understand how the temperature ramp/soak before graphene growth affects the composition and microstructure of the film, we measured the composition, microstructure, and crystal structure of the film at two compositions as deposited and after the heat treatment (60 min at 1000 °C). Figure 2 shows the SEM images (a)–(d), EDS maps (e)–(h), and X-ray diffraction data of the (i) intensity and (j) normalized intensity of two film regions before and after heat treatment at the nearly equiatomic concentration and as will be discussed below, near the optimum bilayer growth composition. Consistent with Figure 1, the as-deposited grains were nanogranular (sub 100 nm) (see inset in Figure 2a,b) for higher resolution images. EDS analysis revealed an as-deposited composition of Cu_51_Ni_49_ and Cu_68_Ni_32_ at the two locations that were evaluated. The EDS maps confirm the even distribution of the Cu and Ni expected from the solid solutions. The annealed films underwent recrystallization and severe grain growth, with grain sizes approaching 10 μm, and thus these larger grains promote larger area catalysts for graphene growth. EDS analysis of the composition of the comparable annealed region revealed compositions of Cu_54_Ni_46_ and Cu_65_Ni_35_, both in good agreement with the as-deposited films and within reasonable error of the specific location and technique. The XRD patterns all showed a preferred (111) orientation (with a small (222) reflection at higher angles—not shown) and, as expected, the diffraction intensity of the as-deposited films were much lower and broader than the annealed films. The two compositions had the expected peak shifts associated with the Cu*_x_*Ni_1−*x*_ solid solution, where higher Cu led to a shift to lower 2θ. After heat treatment, the peaks narrowed, and the intensity grew consistent with the observed recrystallization and grain growth demonstrated in the SEM images. Another peak emerges in the annealed films at approximately 51 degrees, which are consistent with a CuNi-silicide and/or Ni-silicide which result from reactions with the substrate. Comparing the as-deposited and annealed films, there was also a shift to higher 2θ after annealing, which was consistent with either relief of compressive stress in the films. The as-deposited near equiatomic film had a lower intensity in both the as-deposited and annealed film, which was consistent with a decrease in the melting point at a higher Cu in the solid solution. Consistent with this observation, the observed grain size distribution was also slightly higher in the annealed SEM image of the higher Cu content film.

As the graphene growth is at a high temperature and the total high-temperature exposure time is on the order of ninety minutes, the question arises if this exposure affects the induced gradient. As an estimate of the diffusion distance, we used the Cu self-diffusion coefficient (D_Cu_) at 1000 °C which is approximately 1 × 10^−13^ m^2^/s [25]. An estimation of the mean displacement distance (*x*) was given by *x* = (4Dt)^0.5^. Thus, during the total high-temperature exposure, this suggested a mean displacement distance of approximately 465 μm. Thus, as realized by our EDS results in Figure 2, the 1000 °C 90 min exposure had little effect on the as-deposited concentration profile.

### 3.2. Discovering Optimum Cu at.% in CuNi Alloy for Bilayer CVD Graphene

Subsequent to characterizing the gradient catalyst film, a graphene film was CVD grown and characterized via Raman spectroscopy. The initial film was grown with a 0.07 L/min flow rate of the 5% CH_4_ in Ar. Graphene Raman spectra on the catalyst were taken at various Cu compositions is illustrated (Figure 3). Figure 3b shows the magnified 2D peak for the graphene. At a high Cu concentration (>68% Cu), the 2D peak fitting can be accurately described with a single Lorentzian peak which is centered at ~2681 cm^−1^. This behavior is consistent with monolayer graphene [26]. As the Cu in the catalyst decreased in concentration below 68% Cu, the 2D peak exhibits broadening and can no longer be fit with a single Lorentzian peak. Instead, the 2D peak was composed of two peaks centered at approximately 2670 and 2715 cm^−1^. This behavior is indicative of multilayer graphene. Therefore, there is some content of multilayer in the graphene grown on the catalyst with a Cu concentration of less than 68%. Furthermore, the 2D/G ratio can also provide information about the graphene thickness (Figure 3c). The 2D/G ratio increases with increasing Cu concentration. A 2D/G ratio of >1 can be expected for monolayer graphene; bilayer graphene exhibits a 2D/G ratio of ~1; and graphene with ≥3 layers typically exhibit a 2D/G ratio of <1. Therefore, at Cu compositions above ~68%, the resultant graphene is predominately monolayer. For Cu composition below ~39%, the graphene is predominantly > 3 layers. For Cu compositions 39% < *x* < 68%, there is a transition from monolayer to bilayer to multilayer graphene.

To better study this interesting monolayer–bilayer–multilayer transition region, we sputtered a new catalyst sample with approximately twice the copper deposition rate that yielded a Cu-rich catalyst with a smaller concentration gradient (42–94 at.% Cu) and also increased the 5% CH_4_ in Ar flow rate to 0.25 L/min. This new gradient catalyst had a smaller composition gradient which allowed the transition region to be more finely and accurately studied. The increased CH_4_ flow rate should have the effect of increasing the carbon content solubilized in the catalyst when not fully saturated, and hence increase the graphene film thickness at comparable catalyst compositions.

Raman spectra of graphene on the catalyst as a function of the Cu catalyst concentration in the smaller concentration gradient wafer are illustrated (Figure 4a). Figure 4b reports the magnified 2D Raman peak as a function of Cu composition. At high Cu concentrations (>69%), the resulting graphene Raman spectra can be fitted with a single Lorentzian peak centered at ~2688 cm^−1^, consistent with monolayer graphene. As the Cu concentration is decreased (<69%), multiple peaks emerge, centered at 2668 and 2710 cm^−1^, which are consistent with the emergence of bilayer and multi-layer graphene. Clearly, the 2D/G ratio (Figure 4c) again decreases with decreasing Cu concentration, which suggests the layer number is increasing [26]. Once again, a graphene 2D/G ratio < 1 was exhibited by graphene grown with a low Cu concentration (<51%), which indicated the presence of a multilayer. At a high Cu concentration (>85%), the 2D/G ratio was >1 and indicated the presence of monolayer graphene. A transition region from monolayer to multilayer graphene was seen at Cu compositions 51% < *x* < 85%. 

### 3.3. Confirmation of Variation in Graphene Thickness with Gradient in Composition of CuNi Alloy

To further validate the Raman analysis, we transferred the graphene film from the catalyst substrate to a bare silicon wafer and observed representative regions via optical and Raman spectroscopy. Graphene Raman spectra on a SiO_2_/Si wafer are easier to analyze than on the CuNi catalyst because the catalyst can exhibit a luminescent background during Raman acquisition. Optical images (Figure 5, top row) and Raman 2D/G ratio maps (Figure 5, bottom row) are reported of the graphene films grown on 42 at.% Cu, 51 at.% Cu, 61 at.% Cu, 69 at.% Cu, 85 at.% Cu, and 94 at.% Cu catalyst composition. 

The optical images and 2D/G Raman maps showed that the high Ni content catalyst results in non-uniform, multilayer graphene growth with much of the region having a 2D/G < 1. Note that high Ni catalyst (<51% Cu) concentration typically results in large variations in graphene thickness and non-uniformity. Furthermore, graphene is often thicker at grain boundaries because these regions have high carbon mobility and results in more graphene precipitation during cooling [21,22]. As the gradient alloy approached a concentration of 61% Cu, the resulting graphene is made up of predominantly bi-layer coverage, which is confirmed by a 2D/G ratio of ~1. Progressing to the 69 at.% Cu region, this region exhibits a single-to-bilayer transition region and we observe about half of this region is single layer and half is bi-layer. Finally, at catalyst concentrations >85 at.% Cu (Figure 5), the graphene is nearly all single layer. Dark regions in the 2D/G ratio maps in this region mostly correspond to small rips in the graphene from the transfer process. 

## 4. Discussion

To demonstrate the utility of the rapid materials discovery process, we subsequently grew a full wafer with a copper composition of ~63 at.% and transferred the large-area graphene layer to a silicon wafer. A low magnification optical micrograph of a graphene film grown was shown (Figure 6a, left). The higher magnification image illustrated regions that are single layer, bilayer, and trilayer, where clearly the bilayer area is the largest area fraction (Figure 6a, right). A representative Raman spectrum from the 2L region of the transferred graphene which confirms the dominant bilayer graphene as the 2D/G ratio is close to 1 is reported (Figure 6b). A quantitative overview of the areal coverage as a function of the number of layers of graphene is shown in Figure 6c. The bilayer coverage makes up >70% of the area of the graphene film, with portions of multilayer accounting for the majority of the rest of the coverage.

## 5. Conclusions

The number of graphene layers grown in a CVD process is typically tuned by the growth catalyst as well as growth parameters such as temperature, gases, and duration of the growth process. We were able to fine-tune the number of graphene layers produced by combinatorically sputtering a thin film of Cu_x_Ni_1-x_ alloy on Si/SiO_2_ substrate. This approach produced a thin-film Cu*_x_*Ni_1−*x*_ alloy catalyst with a gradient in composition from 25 at.% < *x* < 83 at.% across the 100 mm diameter substrate (~0.87%/mm composition gradient). While keeping growth conditions constant, the graphene synthesized on the gradient catalyst ranged from monolayer graphene on the Cu-rich side, to multilayer graphene on the Ni-rich side. A composition of ~61 at.% Cu yielded predominantly bi-layer coverage. A wafer-scale Cu*_x_*Ni_1−*x*_ alloy was grown with uniform ~63 at.% Cu concentration using the co-sputtering technique. This catalyst yielded wafer-scale CVD graphene with >70% coverage being bi-layer graphene. The combinatorial sputtering approach provides a technique to rapidly study alloy catalyst composition for CVD processes. Additionally, the combinatorial sputtering approach can be used to study catalysts for CVD processes beyond graphene, such as carbon nanotubes and hexagonal boron nitride.

## Figures and Tables

**Figure 1 nanomaterials-12-01553-f001:**
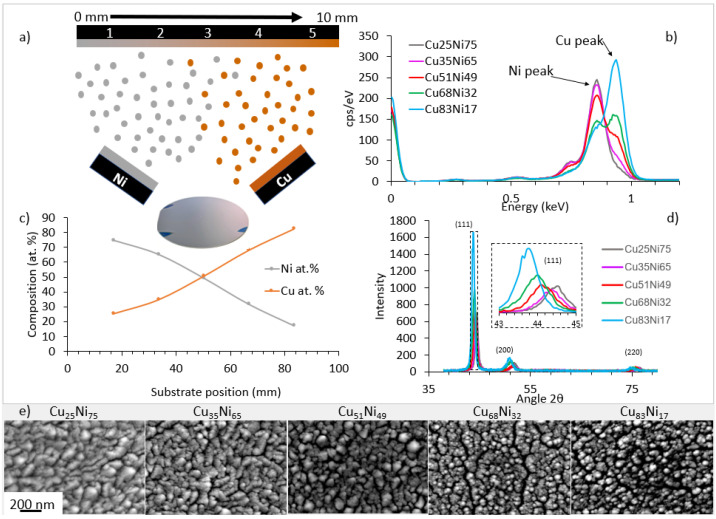
(**a**) Schematic illustrating the combinatorial Cu*_x_*Ni_1−*x*_ thin-film sputtering, (**b**) EDS spectra, (**c**) resultant Cu and Ni concentration across the 5 positions. Inset in figure (**c**) is a photograph of a full as-deposited wafer illustrating the gradient increasing copper from left to right. (**d**) XRD plot showing variation in intensity versus 2θ for the 5 Cu*_x_*Ni_1−*x*_ positions measured; note the inset of the (111) peak shows a clear shift to higher 2θ with increasing Ni, consistent with an expansion of the lattice parameter of the Cu*_x_*Ni_1−*x*_ solid solution. (**e**) SEM micrographs of the as-deposited catalyst Cu*_x_*Ni_1−*x*_ thin-film catalyst.

**Figure 2 nanomaterials-12-01553-f002:**
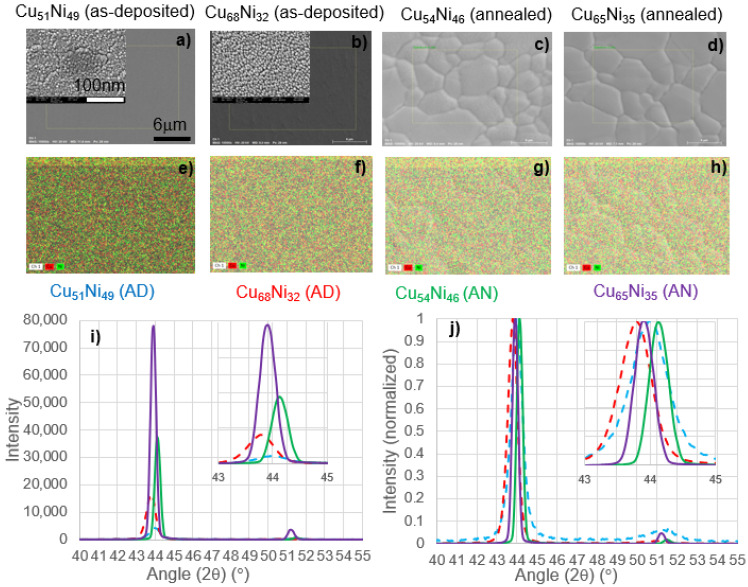
SEM (**a**–**d**) and EDS maps (**e**–**h**) and XRD data (**i**), (**j**) of an as-deposited and annealed film region near the equiatomic composition as well as near the optimum bilayer graphene growth composition as will be discussed below.

**Figure 3 nanomaterials-12-01553-f003:**
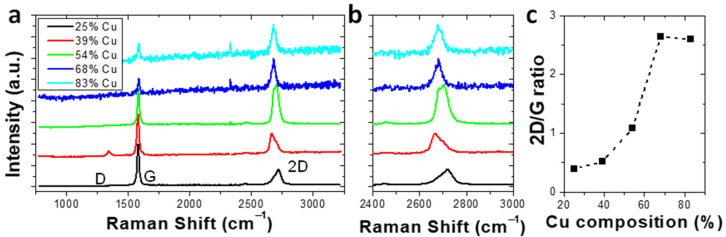
(**a**) Representative Raman spectra of graphene layers grown on Cu*_x_*Ni_1−*x*_ catalyst at regions with varying Ni concentrations. (**b**) Raman spectra of the graphene 2D peak. (**c**) 2D/G ratio of Raman spectra as a function of catalyst Cu composition.

**Figure 4 nanomaterials-12-01553-f004:**
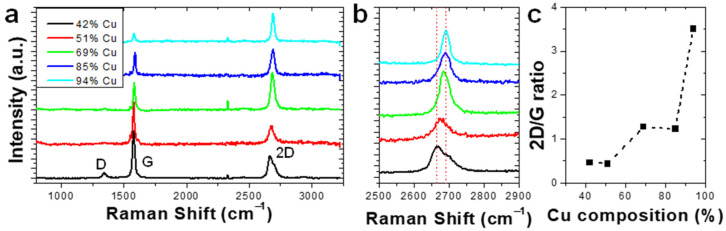
(**a**) Raman spectra of graphene grown on Cu*_x_*Ni_1−*x*_ catalyst at regions with varying Ni concentrations. (**b**) Raman spectra of the graphene 2D peak. (**c**) 2D/G ratio of Raman spectra as a function of catalyst Cu composition.

**Figure 5 nanomaterials-12-01553-f005:**
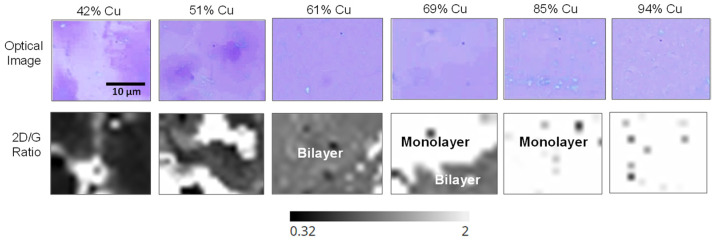
Optical images of graphene were transferred to a Si wafer which was grown on the Cu*_x_*Ni_1−*x*_ gradient catalyst (42–94 at.% Cu) (**top row**). Raman maps report the 2D/G ratio of the transferred graphene. Maps correspond to the optical images (**bottom row**).

**Figure 6 nanomaterials-12-01553-f006:**
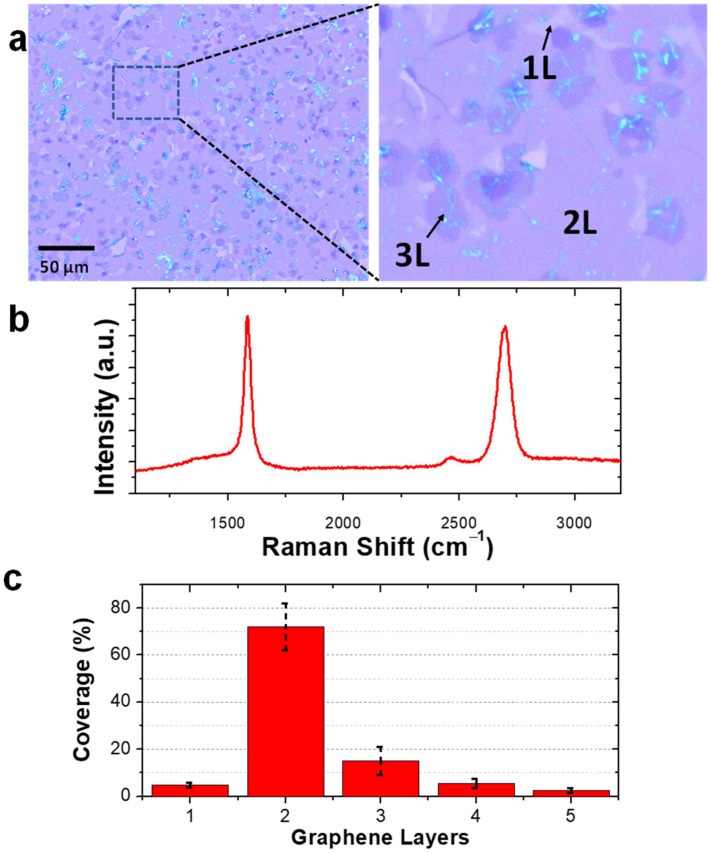
Large area bilayer graphene coverage demonstration (**a**) optical images with blow-up showing predominantly 2 L coverage with small islands of 1 L and 3 L domains, (**b**) plot of Raman spectra with I_2D/G_ ≅ 1 elucidating 2 L coverage, (**c**) bar graph showing ≥70% coverage of bilayer graphene growth with small amounts of monolayer and multilayer (≥3) domains.

## Data Availability

Data available by request.

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
