# Peer review of "Combinatorial Cu-Ni Alloy Thin-Film Catalysts for Layer Number Control in Chemical Vapor-Deposited Graphene"

_nanomaterials, 2022, doi:10.3390/nano12091553_

Round 1

Reviewer 1 Report

The authors addressed my concerns in the revised manuscript, thus it is now qualified for publication after the figure numbers are corrected.

Author Response

We would like to thank the reviewers for their helpful comments regarding our manuscript.  Below are the original referee comments in bold text and our responses in red.  We have appropriately updated the manuscript to answer the requested edits, and the manuscript is much-improved and we believe now suitable for publication in the Nanomaterials. 

Referee 1

The authors addressed my concerns in the revised manuscript, thus it is now qualified for publication after the figure numbers are corrected.

We thank the reviewer for the positive response and we have fixed figure numbers. 

Reviewer 2 Report

Title: Combinatorial Cu-Ni alloy thin film catalysts for layer number control in chemical vapor deposited graphene

Journal: Nanomaterials

Ms.  No.: nanomaterials-1695229

Khanna et al. demonstrated the synthesis of CVD graphene using combinatorial Cu-Ni alloy catalysts, allowing the control of the number of graphene layers. The Authors have shown that depending on the CuxNi1-x film content single, bi-, or multi-layer graphene growth occurs. The catalyst layers are characterized using XRD, SEM, and EDX, while graphene layers were mostly characterized using Raman spectroscopy. The work is interesting and can be helpful for the selective synthesis of bilayer graphene. The work has already been reviewed and the Authors provided extensive responses and modified the manuscript according to Reviewers’ comments. I believe that the major issues are resolved, while the manuscript still needs a minor polishing before publication.

For example, some figure captions should be modified:

  1. “Figure 2. Representative Raman spectra grown on CuxNi1-x catalyst at regions with varying Ni concentrations.” Should be “Representative Raman spectra of graphene layers grown…” or similar.
  2. More careful formatting is needed in the newly added parts of the revised manuscript. In several places, subscripts are missing, while in lines 217-218 reference is missing (“at 1000oC which is approximately 1x10-13 m2/s [ref ].”)
  3. Section “3.2 Discovering optimum Cu at.% in CuNi alloy for bilayer CVD graphene”. I find it a crucial point of the work, but this section is quite short and the Authors have just went through the findings without proper quantification of their results and deeper analysis of Raman spectra. Even some statements should be more precise, like “At high Cu concentrations, the resulting graphene can be fit with a single Lorentzian peak” – Raman spectra are fitted, not graphene. I believe that the Authors should extend the analysis of Raman spectra and really prove to possible readers that they have single, bi-, or multilayer graphene. I am sure that the Authors will easily resolve this issue and provide a more detailed analysis in this section.
  4. Figure 5 (a). The scale bar is missing from the image on the right. We might guess it based on the scale bar on the left image, but such suggesting is better not to be done.

Author Response

We would like to thank the reviewers for their helpful comments regarding our manuscript. Please find the response in the attachment.

This manuscript is a resubmission of an earlier submission. The following is a list of the peer review reports and author responses from that submission.

Round 1

Reviewer 1 Report

The manuscript reports the CVD graphene synthesis on co-sputtered Cu-Ni alloy film, in which the graphene layer number could be adjusted by the Cu or Ni composition. As the Ni content increases, the graphene transforms from monolayer, bilayer to multilayer. As an example, the graphene with bilayer coverage > 70 % was achieved on a uniform alloy film with a copper composition of ~ 63 at.%. The result is reasonable and predictable due to the different carbon solubility between Cu and Ni, and exhibits little novelty compared with the previously published work. In addition, the morphology and crystalline facets of the alloy film before graphene growth is provided in figure 1, and the graphene films after growth are also shown. However, the morphology and grains facets of the alloy film after graphene growth are omitted. Since the alloy substrate went through temperature ramping and annealing (1000 ℃ for 1 h according to the Methods), the Cu and Ni interdiffusion and film morphology evolution could be very severe in this process, the graphene layer number change could not be solely correlated with the composition of the alloy film as a result. So in my opinion, the characterizations of the alloy film after graphene growth should be provided as well.

Reviewer 2 Report

Comments

  • How are the Cu and Ni distributed along the thickness direction at a specific position on the CuxNi1–x catalyst wafer? Homogeneously or with a composition gradient forming just as the case along the diameter direction? More experimental proofs are suggested to be provided.
  • Considering the relatively high moving abilities of Cu and Ni atoms under such a high temperature (1000 ℃), which might homogenize the composition distribution of the Cu/Ni catalyst film via thermal diffusion driven by the composition gradient, would the 1-h thermal annealing under H2 and Ar before graphene growth exert any influence on the composition and its gradient? More experimental results is suggested to be displayed.
  • Surface roughness of the catalyst or substrate largely influences growth behaviors of graphene. The authors are suggested to give some characterization results regarding the surface roughness of the CuxNi1–x catalyst film before and after 1-h thermal annealing in Ar and H2 atmosphere, such as AFM images (along with corresponding height line profile), or cross-section SEM images if the surface is very coarse.
  • It has been noted that the Raman spectra of graphene grown on the Cu/Ni films with approximate Cu compositions in Figure 2 (g., 51% Cu) and Figure 3 (e.g., 54% Cu) differs a lot. What’s the cause of such difference?
  • The authors are suggested to supplement experimental details regarding graphene transfer process (mentioned in Section 3.3) in the Materials and Methods section. By the way, the supporting information of this manuscript seems inaccessible, the authors are suggested to upload it if it was indeed missed.